# The Influence of Tree Species on the Recovery of Forest Soils from Acidification in Lower Saxony, Germany

Bernd Ahrends *, Heike Fortmann and Henning Meesenburg

Department of Environmental Control, Northwest German Forest Research Institute (NW-FVA),
D-37079 Göttingen, Germany; heike.fortmann@nw-fva.de (H.F.); henning.meesenburg@nw-fva.de (H.M.)
* Correspondence: bernd.ahrends@nw-fva.de; Tel.: +49-551-69401202

**Abstract:** Atmospheric acid deposition has increased sharply since the beginning of industrialization but has decreased considerably since the 1980s owing to clean-air policies. Soil acidification induced by an input of acidity has been demonstrated in numerous studies using repeated forest-soil inventories. So far, relatively few data have been sampled to analyze long-term soil trends and only a few studies show the recovery of forest soils from acidification, whereas the recovery of surface waters following declining acid deposition is a widespread phenomenon. To assess a possible recovery from acid deposition, soil resampling data from 21 forested permanent soil-monitoring sites in Lower Saxony (Germany) were evaluated. For most sites, at least three repetitions of inventories from a period of 30 to 50 years were available. Trend analyses of indicators for the acid-base status of unlimed forest soils using generalized additive mixed models (GAMM) show either a trend reversal or a stagnation of the acid-base status at a strong acidification level. The recovery, if indicated by an increase of soil pH and base saturation, of soils from plots with deciduous trees appears to have occurred faster than in coniferous forest stands. This observation may be attributed to a larger amount of temporarily stored sulfur in the soil because of the higher atmospheric input into coniferous forests. As indicators for the acid-base status still show considerable soil acidification, mitigation measures such as forest liming still appear to be necessary for accelerating the regeneration process.

**Keywords:** base saturation; repeated soil sampling; acidification; forest soils; recovery; sulfur deposition; generalized additive mixed models





## 1. Introduction

The Industrial Revolution in Europe resulted in the increasing emission of acidifying pollutants into the atmosphere. The deposition of acidifying and eutrophying substances drastically altered the stability, nutrient cycles, and growth of forest ecosystems for several decades [1–3]. Sulfur (S) was the major component of acid deposition since the beginning of industrialization until the 1980s. The strength of the soil acidification dynamics because of acid input is primarily determined by the ability of the soil to buffer the input of acids [4,5]. The buffer capacity of soils increases with carbonate and clay mineral contents and is lowest in sandy soils. In Lower Saxony (Germany), this is particularly the case in the forested regions of the lowlands and at Harz, Solling, and Hils mountains. As a consequence of acid deposition, forest soils experienced a severe loss of base cations (Bc: $Ca^{2+}$, $Mg^{2+}$, $K^+$) with seepage water. Significantly acidified soils with a low buffer capacity show a decline in soil pH and base saturation. A loss of base cations may lead to nutrient imbalances at base-poor sites [2]. For example, at the Harz and Solling mountains in Lower Saxony, the high sulfur deposition caused severe soil acidification and Mg-deficiency symptoms of the forest stands, and large areas were affected by forest decline [6–8]. Beginning in the 1980s, clean-air policies resulted in a considerable decrease of sulfur deposition in Europe, which continues until today [9–11]. The dynamic development of sulfur deposition raises questions about the recovery of these soils from acidification [12]. This is especially true

for forest ecosystems that have not yet been limed. The adsorption and release of sulfur in forest soils play decisive roles in the recovery of forest soils from acidification [13,14]. For predictions of recovery, atmospheric inputs, soil characteristics, soil S pools, and their dynamics have to be considered.

The deposition of oxidized and reduced nitrogen (N) also contributes to the acidification of forest soils [15]. However, the reductions in N deposition recorded since the early 1990s have been less pronounced than those for sulfur [9] and, accordingly, the acidification of Europe's forest soils shows a limited response to the decrease in N deposition since the 1990s [16].

From the comparison of coniferous and deciduous forest stands at comparable sites, it can be inferred that deciduous forests receive less sulfur via total deposition [17]. For the deposition process, the structure of the canopy plays an important role [18,19]. Therefore, lower deposition rates can be expected for the forest functional types that lose their leaves in autumn. In many cases, decreases in sulfur deposition have been linked directly to the degree of recovery of forest soils from acidification [20]. Accordingly, it can be assumed that because of larger stored S pools from higher sulfur deposition inputs, the recovery in conifers is significantly delayed. A long-term study at European beech and Norway spruce ecosystems in the Solling area, Germany, shows some indications of recovery of base cation to aluminum (Bc/Al) ratios in mineral soils in the beech site, whereas no recovery is observable in the spruce site [21].

The recovery of forest soils from acidification is an important topic for environmental policy with respect to future emission-reduction goals. For forest management, the planning of liming programs must be considered in view of natural regeneration. However, indications of the recovery of forest soils, including soil solution, from acidification are only sparse [20,22–24], whereas the recovery of surface waters is well documented [25–28]. In the studies of Berger et al. [29] and Reininger et al. [30], there were no indications of a recovery from acidification, or even an ongoing acidification process. Cools and De Vos [31] found a statistically significant change in the soil pH(CaCl$_2$) and base saturation of European forest soils between 1994 and 2006. However, the detection of long-term trends was very different depending on the soil type, soil depth, and acid-base status of the soil.

The verification of changes from subsequent soil inventories is very difficult owing a high small-scale variability in the forest floor and the mineral soil [32–34], the low rate of change of soil properties, and the non-linearity of trends [35,36]. There are still some doubts as to whether repeated soil sampling is an efficient tool to distinguish between temporal and spatial variability [37]. Previously, soil changes were assumed to take place over timescales of centuries to millennia [22]. To detect such changes, chronosequences were frequently used to analyze a temporal development [38,39]. The high rate of soil changes after the strong increase of sulfur deposition between 1950 and 1970 offered the possibility to detect changes with repeated inventories, despite several uncertainties [21,40–42]. In contrast, the recovery of forest soils from acidification after strong reduction of the acid load appears to be delayed because of the gradual release of temporarily adsorbed sulfur from soil pools [29].

The evidence of the recovery of forest soils from acidification with repeated soil inventories at different sites also poses several statistical challenges, as most data are from observational studies rather than from factorial experiments. Due to the experimental design and the use of soil chemical response variables, the assumptions of the parametric statistical methods are not always fulfilled. Generalized additive mixed models (GAMMs) provide an approach that allows (1) the analysis of response variables with non-Gaussian distributions, such as base saturation; (2) the inclusion of random effects to account the "pseudo replicated" structure of the data (correlated errors among inventories on the same sampling site); (3) the consideration of non-linear processes; (4) to account for sampling heterogeneity across space and time; and (5) inconsistencies in the timing of soil inventories that could be addressed by modelling the within-year-variation as a covariate [43–45].

The objective of our study was to improve the detection and understanding of long-term changes of acidification of highly polluted forest soils in Lower Saxony, Germany. Specifically, we considered two main research questions. First, are there significant indications of forest soil recovery after three decades of reduced sulfur deposition? Second, are the trends of the indicators for soil acidification different for tree species owing to the lower atmospheric sulfur input in deciduous forest stands? The pH values, either measured in water (pH(H$_2$O)) or in calcium chloride solution (pH(CaCl$_2$)) and base saturation, were used as indicators for the acid-base status of the forest soils in this study.

## 2. Materials and Methods

### 2.1. Study Sites

Included in this study were all 21 forest study plots (Figure 1) with terrestrial soils of the network of permanent soil monitoring plots in Lower Saxony (in German: BDF = Bodendauerbeobachtungsflächen [46]) existing since 1992. Some of the plots belong also to the ICP Forests Intensive Monitoring Programme Level II established under the UNECE Convention on Long-Range Transboundary Air Pollution [47].

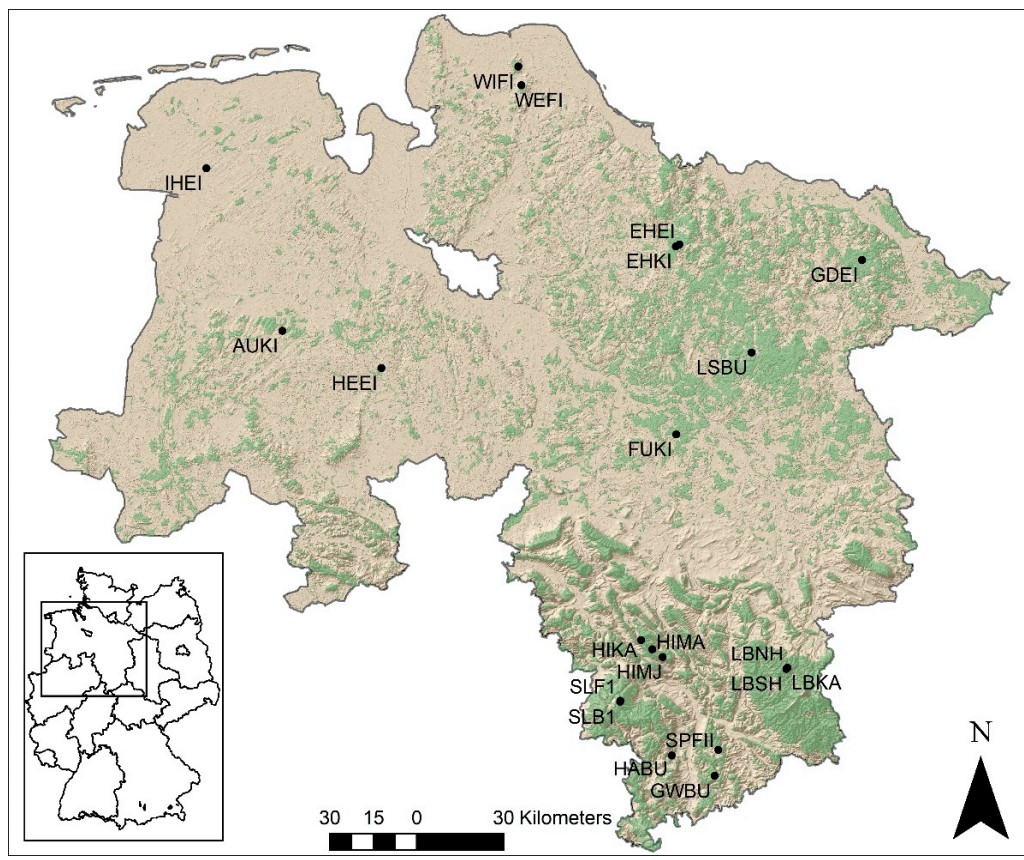

**Figure 1.** Location of 21 permanent soil-monitoring plots in Lower Saxony, Germany. The forested areas are represented with the color green. Abbreviations show the last four letters of the plot code (see Table 1).

The most frequent tree species is the Norway spruce (n = 10) (*Picea abies* (L.) H. Karst), followed by European beech (n = 4) (*Fagus sylvatica* L.), oak (n = 4) (*Quercus robur* L. and *Qu. petraea* (Matt.) Liebl.), and Scots pine (n = 3) (*Pinus sylvestris* L.). The sites are located at altitudes between 31 and 657 m a.s.l. Mean air temperature and mean annual precipitation (1981–2010) ranged from 6.4 to 9.7 °C and from 684 to 1443 mm, respectively. Additional information on the sites is provided in Table 1. The most frequent soil type is Podzol (n = 13), followed by Cambisol (n = 4), Luvisol (n = 2), Fluvisol (n = 1), and Planosol (n = 1).

**Table 1.** Location and plot characteristics for repeated soil-sampling plots in Lower Saxony.

| Location | Plot Code | Lat [°] | Long [°] | MTS [-] | Alt [m] | Slope [°] | Aspect [°] | Stype [WRB] | MAT [°C] | MAP [mm] |
|---|---|---|---|---|---|---|---|---|---|---|
| Westerberg | F001WEFI | 53.67 | 9.09 | spruce | 37 | 1.1 | 311 | Podzol | 8.7 | 903 |
| Ehrhorn | F002EHEI | 53.18 | 9.90 | oak | 110 | 2.9 | 337 | Cambisol | 8.5 | 843 |
|  | F012EHKI | 53.17 | 9.88 | pine | 82 | 1.1 | 241 | Podzol | 8.8 | 826 |
| Lüss | F003LSBU | 52.84 | 10.17 | beech | 115 | 2.3 | 310 | Podzol | 8.5 | 859 |
| Fuhrberg | F004FUKI | 52.59 | 9.87 | pine | 37 | 0.1 | 107 | Podzol | 9.7 | 684 |
|  | F005LBNH | 51.86 | 10.42 | spruce | 613 | 15.8 | 19 | Podzol | 6.5 | 1453 |
| Lange Bramke | F022LBSH | 51.86 | 10.41 | spruce | 603 | 23.0 | 151 | Podzol | 6.4 | 1446 |
|  | F023LBKA | 51.86 | 10.42 | spruce | 656 | 8.1 | 109 | Podzol | 6.5 | 1453 |
| Solling | F006SLB1 | 51.76 | 9.58 | beech | 506 | 1.9 | 207 | Cambisol | 7.0 | 1246 |
|  | F007SLF1 | 51.76 | 9.58 | spruce | 507 | 1.4 | 92 | Podzol | 7.0 | 1245 |
| Harste | F008HABU | 51.59 | 9.83 | beech | 250 | 5.5 | 82 | Luvisol | 8.5 | 774 |
| Göttingen | F009GWBU | 51.53 | 10.05 | beech | 421 | 0.3 | 246 | Luvisol | 7.5 | 897 |
| Wingst | F010WIFI | 53.73 | 9.07 | spruce | 34 | 3.7 | 102 | Podzol | 8.7 | 933 |
| Ihlow | F011IHEI | 53.41 | 7.45 | oak | 53 | 0.3 | 274 | Podzol | 9.4 | 846 |
| Göhrde | F013GDEI | 53.12 | 10.84 | oak | 97 | 1.9 | 127 | Podzol | 8.5 | 742 |
| Heerenholz | F014HEEI | 52.80 | 8.37 | oak | 48 | 0.3 | 204 | Fluvisol | 9.4 | 795 |
|  | F016HIKA | 51.95 | 9.69 | spruce | 424 | 24.3 | 253 | Podzol | 7.6 | 1196 |
| Hils | F017HIMA | 51.92 | 9.74 | spruce | 253 | 6.3 | 244 | Cambisol | 8.2 | 1020 |
|  | F019HIMJ | 51.90 | 9.79 | spruce | 317 | 14.0 | 26 | Cambisol | 7.8 | 1075 |
| Spanbeck | F020SPFI | 51.61 | 10.07 | spruce | 251 | 1.7 | 72 | Planosol | 8.6 | 830 |
| Augustendorf | F021AUKI | 52.91 | 7.86 | pine | 31 | 1.0 | 102 | Podzol | 9.4 | 843 |

Lat: latitude, Long: longitude; MTS: main tree species, Alt: altitude above sea level, Stype: soil type, MAT: mean air temperature, MAP: annual mean precipitation (reference period 1981–2010).

### 2.2. Estimation of Total Sulfur Deposition

Deposition assessments at some of the permanent monitoring plots were conducted according to the ICP Forests Manual on the sampling and analysis of deposition [48]. Fifteen continuously open bulk samplers were placed under the forest canopy. At plots with European beech, stemflow was assessed at a subset of the trees [48]. Samples were collected at least fortnightly, filtered, and stored in the dark at about 4 °C before being chemically analyzed.

As already mentioned in the introduction, we focused on sulfur deposition because the acidification of Europe's forest soils shows a limited response to the decrease in N deposition since the 1990s [16]. While sulfur deposition fluxes are only observed at a subset (9 of 21) of the permanent inventory plots, we estimated the modeled deposition fluxes of sulfur from the mapped atmospheric deposition data for Germany that were provided by the Federal Environment Agency (UBA) [49]. These data only cover the period from 2000–2015. To obtain the sulfur deposition time series from 1950 to 2020 for each monitoring site, we adapted temporal reconstruction methods. This was done with a simplified version of the model MAKEDEP by Alveteg et al. [50]. For the scaling of the sulfur deposition, we distinguished between marine and non-marine deposition. For sea spray, we assumed a constant deposition over time. For the non-marine proportion, we created a standard function for scaling. This function was based on the trend of sulfur deposition for Europe described by Engardt et al. [9]. This curve was adjusted using observed deposition data from sites in Lower Saxony. Lower Saxony received very high sulfur deposition inputs in some regions in the past, especially in the Solling and Harz mountains. More details on the methods for computing yearly deposition fluxes and the applied scaling function are given in Figure A1.

The quality of regionalized data and the reconstruction process with the same standard scaling function for all sites in Lower Saxony was evaluated at the sites with observed total S depositions (Figure 2). When comparing the observed and estimated deposition, both the site-specific level and the temporal development of the sulfur deposition is well represented with the estimated unified scaling function. An exception is the site F007SLF1 at Solling, where a systematic underestimation of the observed deposition is visible (Figure 2). Therefore, the function was adjusted by increasing the estimated deposition by a factor of 1.45 for this site (see dotted line in Figure 2).

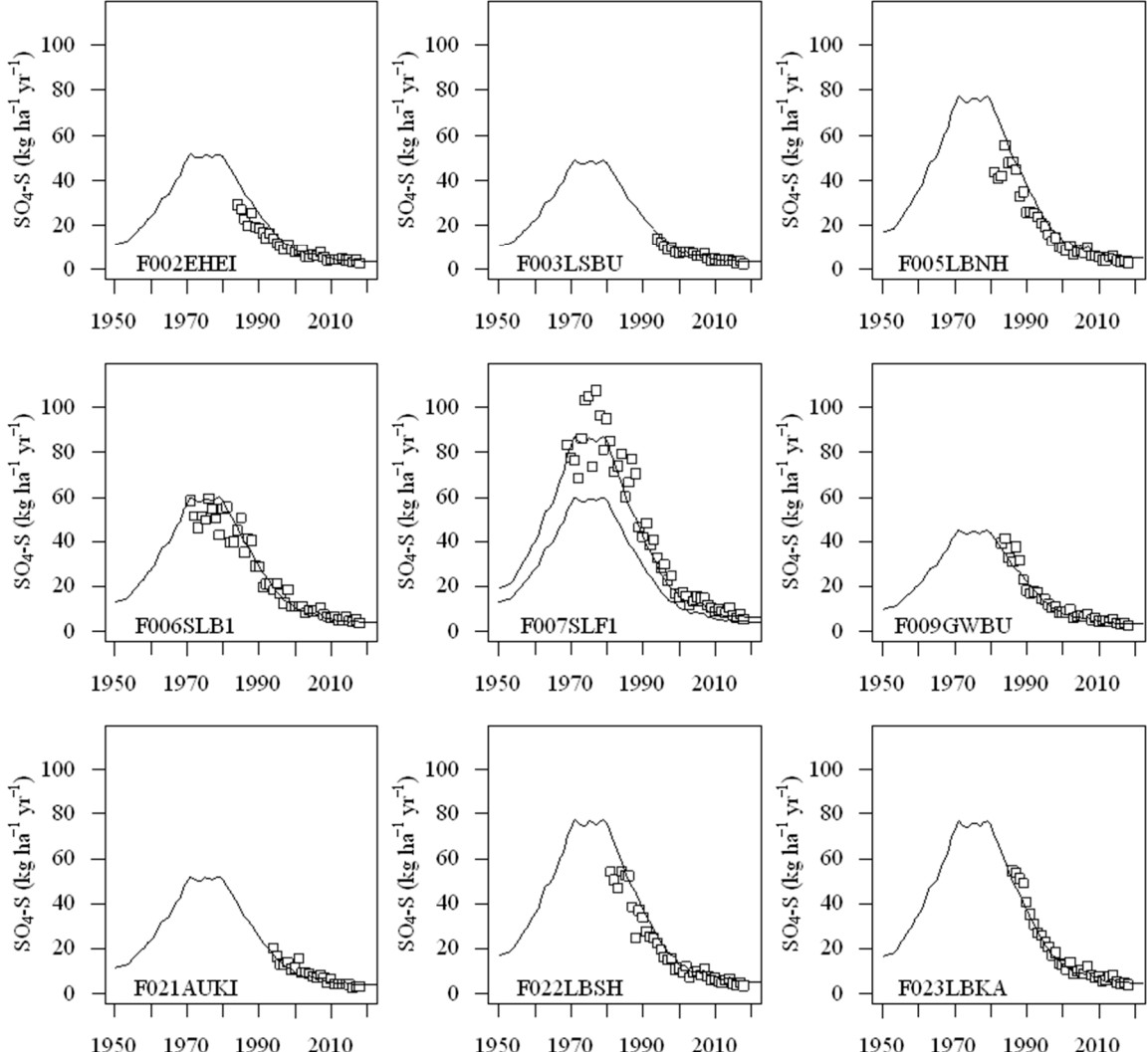

**Figure 2.** The time series of observed total sulfur deposition from throughfall measurements (□) at permanent soil-monitoring plots and the reconstructed time series of sulfur deposition for the period 1950 to 2020. The dotted line shows a site-specific recalibration of the observed data for the F007SLF1 plot.

### 2.3. Sampling Procedures and Chemical Analysis

The sampling of soils at the individual permanent soil-monitoring plots is conducted at intervals of no more than 10 years (Figure 3). At least three inventories are available for most plots (exceptions: F016HIKA, F017HIMA, and F010WIFI). The sampling is conducted in an alternating system with about two inventories each year. The sampling-site design, as well as the sampling procedures, chemical analyses, and quality checks, are documented by Barth et al. [51] and in the ICP Forests Manual on Sampling and Analysis of Soil [52]. Accordingly, soil samples were taken at 24 locations within the plot, which were aggregated

to six composite samples per plot and depth interval. The inventories following this design are described by black circles in Figure 3. The organic layer was divided into a litter (L), a moderately decomposed (Of), and a highly decomposed (Oh) layer, if present. The mineral soil was sampled at fixed depths of 0–5 cm, 5–10 cm, 10–20 cm, and 20–30 cm, and at a maximum 20 cm interval for deeper soil layers. The soil samples were analyzed for their content of the elements carbon (C), N, and phosphorus, for pH, and for their effective cation exchange capacity (CEC). Exchangeable cations ($Na^+$, $K^+$, $Mg^{2+}$, $Ca^{2+}$, $Al^{3+}$, $Fe^{3+}$, $Mn^{2+}$, and $H^+$) were determined after percolating the sieved (<2 mm) soil samples with $NH_4Cl$ and the cations in the percolate were subsequently determined using ICP methods and pH measurements for $H^+$ [53–55]. CEC was calculated as the sum of the cation equivalents [56]. Base saturation (BS) was calculated as a percentage of the exchangeable base cations ($Na^+$, $K^+$, $Ca^{2+}$ and $Mg^{2+}$) from the CEC. For the determination of pH, samples were mixed with a volume ratio of sample to solution of 1:5 with $H_2O$ (pH($H_2O$)) or 0.01 M $CaCl_2$ solution (pH($CaCl_2$)) and pH was determined with a glass electrode. The pH values and base saturation were used as indicators for the acid-base status of the soil. Sample preparation and analysis followed standard procedures [57].

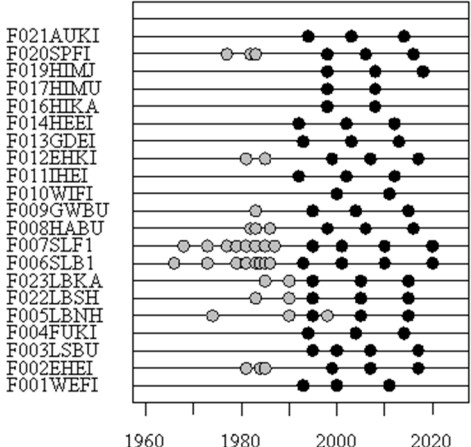

**Figure 3.** Sampling years of soil inventories at the permanent soil monitoring plots in Lower Saxony, Germany. (Black cycles: sampling design complies with the ICP Forests specifications; grey cycles: inventories with partly deviating sampling design, but comparable analytical methods).

For most plots, data from soil inventories from prior to the start of the permanent soil monitoring programme (BDF programme) are available, which allows for the extension of the number of replicates and the study period [14,21,58]. The analytical methods are comparable to the methods used in more recent inventories. The sampling design, however, differs with respect to the number of replicates and sampling depth. These inventories are marked with grey circles in Figure 3. Despite the methodological differences, these data bear important information with respect to the long-term dynamics of the acid-base status.

*2.4. Data Handling*

The average *CEC*, BS, and pH values for the different depth ranges (*d*) of 0–30 cm, 0–50 cm, and 0–100 cm were calculated as follows for *CEC*:

$$CEC_d = \frac{1}{z_d \cdot BD_d} \sum_{l=1}^{n} z_l \cdot BD_l \cdot CEC_l \tag{1}$$

for BS:

$$BS_d = \frac{1}{z_d \cdot BD_d \cdot CEC_d} \sum_{l=1}^{n} z_l \cdot BD_l \cdot CEC_l \cdot BS_l \tag{2}$$

and for pH, the aggregation was done on the basis of the $H^+$ concentration.

The total thickness (depth ranges) d is given by:

$$z_d = \sum_{l=1}^{n} z_l \tag{3}$$

and bulk density (*BD*) by:

$$BD_d = \frac{1}{d} \sum_{l=1}^{n} z_l \cdot BD_l \tag{4}$$

where $z$ = thickness and $l$ = soil layer.

The plots were stratified into sub-groups to enable a more detailed analysis, as well as to consider the different soil chemical processes (such as chemical weathering rates or cation leaching) and other general conditions (such as forest type), as shown in Figure 4.

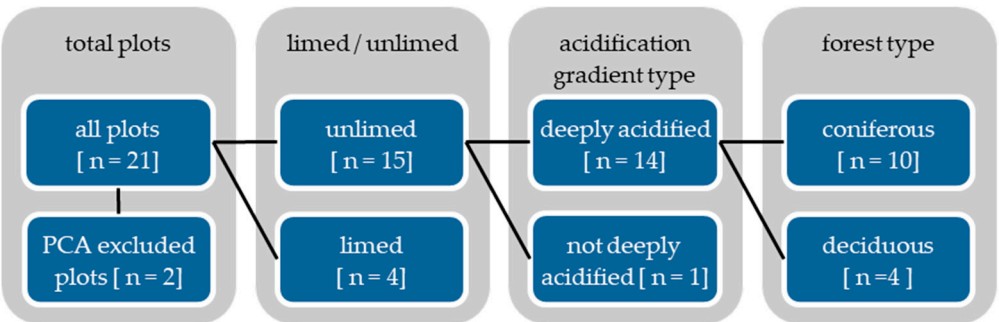

**Figure 4.** Stratification of the inventory plots into functional groups for detailed statistical analysis. Further details on the definition of the "vertical acidification gradient type" can be found in Hartmann and von Wilpert [59] and in Section 3.2. PCA = principal component analysis.

### 2.5. Derivation of Meteorological Data

For a climatic characterization of the study plots (see Table 1), observational data of the German Meteorological Service (Deutscher Wetterdienst, DWD) were used (Table 1). The regionalization of the data from the climate and precipitation stations of the DWD to the soil-monitoring sites was performed using the methods described in Dietrich et al. [60] for the standard period from 1981 to 2010.

### 2.6. Statistical Analysis

R 4.0.3 software (R Development Core Team 2020) was used for the statistical analyses. Depth profile plots and statistics are generated with the package 'aqp': algorithms for quantitative pedology [61].

2.6.1. Detection of "Atypical" Plots Using Principal Component Analysis (PCA)

If some plots are "atypical" (belonging to different clusters) compared to most of the other monitoring plots studied, they tend to bias the interpretation and conclusions of the analysis [62]. PCA can be regarded as a classification procedure and was thus used to detect "atypical" study plots. A given plot belongs to one of the following classes: (a) non-atypical, or (b) atypical [63]. The PCA was performed with the package 'FactoMineR' [64] and the results are displayed using the 'factoextra' package. In addition to the soil chemical acidification indicators (pH, base saturation), the cation exchange capacity (CEC) was also considered as a variable for the PCA. All depth ranges (0–30; 0–50; and 0–100) were used in one analysis to include the effects of the vertical differentiation of the parameters in the soil profiles. With one exception, for all plots the most recent soil inventory was used. At the site F009GWBU, we used the results from the inventory in the year 2004. During this inventory, much greater profile depths were achieved, which makes it easier to compare this site with the others.

2.6.2. Mixed-Effects Models

We used generalized additive mixed models (GAMM) to examine the relationship between the response and the inventory year. Our data consist of $y_{it}$, the base saturation or pH values measured at site i at year t for i = 1, ... ,21 and t = 1966, ... ,2020. Following Knape [45], we separated the smooth trend components from the random among-year variations using the following structure of a mixed effects model:

$$y_{it} \sim \text{Gamma}(\exp(a + S(t) + \varepsilon_t + s_i)), \ \varepsilon_t \sim N\left(0, \sigma^2\right), \tag{5}$$

where a is an intercept, $S(t)$ is a smooth function of the year representing nonlinear changes in base saturation or pH values, $\varepsilon_t$ is a random effect for the year, and $s_i$ is a site effect to account for the among-site variations. With this formulation the temporal change is described as a combination of $S(t)$ and $\varepsilon_t$.

The above formulated model structure from Knape [45] was developed for the count data of populations that often have Poisson-distributed sampling errors. Therefore, a gamma distribution with a log link was used to account for the properties of the soil's chemical variables. We performed checks for the approximate normality of random effects and heteroscedasticity (using the functions 'checkfit' and 'gam.check' [45,65]; for examples, see Figures A2–A4). For assessing unusual observations (outliers), we performed statistical outlier detection with the residual plots and the Grubbs test calculation [66] using the package 'outliers'.

An important choice is the selection of the parameter that determines the flexibility of the smooth functions, since a non-limited smooth component would be able to capture all variations of soil sampling between the different years [45]. If the flexibility is too high there is a risk of overfitting and if it is too low a part of the variation in the data may not be captured well. With an appropriate selection of the degree of smoothing, the $S(t)$ component can be interpreted as the long-term trend while the random effect is capturing the short-term variation, e.g., due to sampling variation/uncertainty. Accordingly, the exact interpretation of the short-term and long-term components depends on the smoothing degree of the long-term component and on how the temporal random effect is modeled. In most cases, we used the automatic selection of smoothing parameters. For subgroups with small sample sizes (e.g., limed sites), it was necessary to reduce the degree of smoothing to avoid temporal overfitting. Standard software to parameterize this type of model is available from the R library 'poptrend' [45]. A major advantage of the function 'ptrend', included in the library 'poptrend', is that significant increases and decreases in the trend could be visualized for different periods of the fitted curve. In conventionally parameterized GAMMs, only the significance of the overall smoothing function can be assessed. More details on methods for computing confidence intervals for the trend estimates and changes are given elsewhere (Figure A1 in [45]).

Uncertainty estimates are an important part of the communication of the trends in soil data. In the function 'ptrend', uncertainty estimates were performed with simulations using parametric bootstrapping based on assuming the asymptotic normality of the estimates [67]. This approach considerably reduced the computational effort relative to fitting GAMs to every bootstrap resample [67], but the cost is that the confidence intervals produced in this way do not account for any uncertainty in the selection of the degree of smoothing [45].

For visualization, the long-term trend ($S(t)$) is standardized. As default in 'ptrend', the long-term trend is standardized with respect to the predicted base saturation or pH value at the first monitoring year. We used for standardization the average predicted base saturation or pH value (*ciBase = mean*) at site i, ignoring the temporal random effects. This has an advantage over the default standardization in that it is less affected by uncertainty in the observed values from the first monitoring year [45].

## 3. Results

### 3.1. Site-Specific Load and Reduction of Sulfur Deposition

For the characterization of the site-specific sulfur deposition load, the estimated annual rates were cumulated from 1950 to 2020 (Figure 5). For the 21 plots considered in this study, the accumulated sulfur deposition in the period between 1950 and 2020 ranged from 1319 to 2819 Mg ha$^{-1}$ with a mean of 1813 Mg ha$^{-1}$ and a median of 1670 Mg ha$^{-1}$. The plot with the maximum deposition load (2819 Mg ha$^{-1}$) is a spruce stand in the Solling area (F007SLF1). The adjacent beech stand (F006SLB1) received an accumulated deposition load of 1936 Mg ha$^{-1}$ of sulfur.

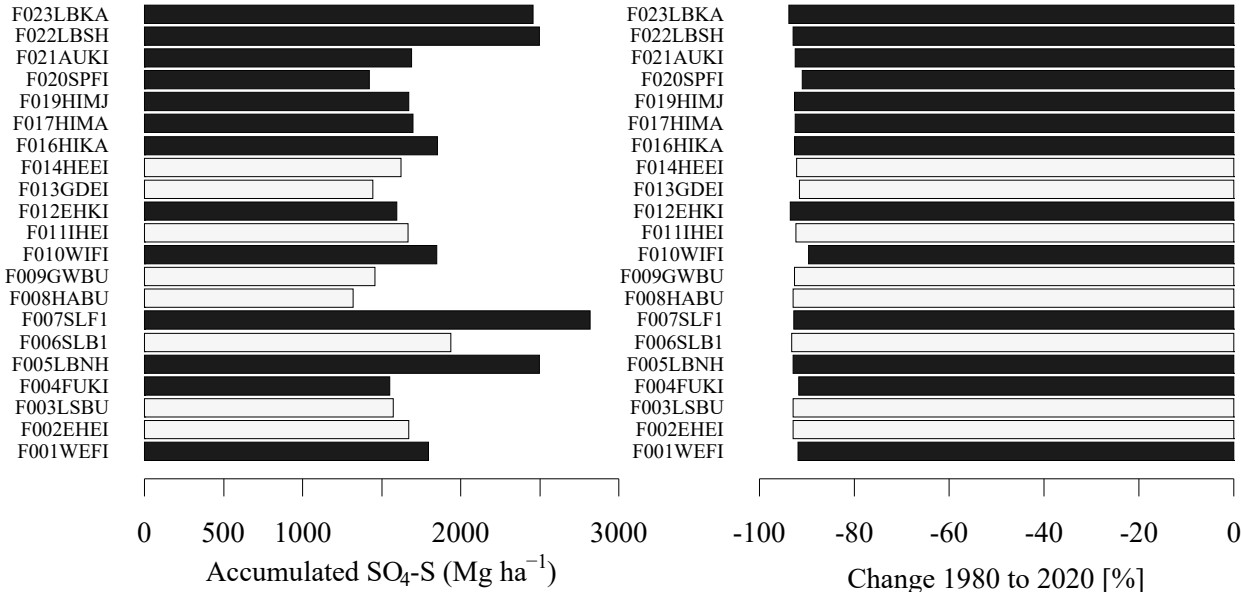

**Figure 5.** Estimated cumulative sulfur load (Σ1950 to 2020) and relative change of sulfur deposition between 1980 and 2020 for the permanent soil-monitoring plots in Lower Saxony, Germany. ■ coniferous forest; □ deciduous forest.

Due to the use of a uniform scaling function in the deposition reconstruction procedure, the relative reduction of the deposition at the individual plots turns out to be similar. The relative changes of the sulfur deposition between 1980 and 2020 ranged from −89 to −94% with a mean of −92%. The lower relative decreases relate to the plots near the coast with a higher sulfur deposition of marine origin.

### 3.2. Soil Chemical Status at the Time of the Last Soil Inventory

At most plots, observed pH(CaCl$_2$) is in the acidic range between pH(CaCl$_2$) 3 and 5 (Figure 6). In the topsoil, pH(CaCl$_2$) is even below 3 at some plots. These soils are predominantly deeply acidified; only at a few plots can higher pH values be found at greater depths (> 70 cm). In contrast, one plot on carbonate bedrock (F009GWBU) shows pH values well above 7 below about 20 cm depth. Forest soils with low base saturation (<20%) in the entire soil profile were observed at most plots, at mountain sites with base-poor silicate bedrock, and at unconsolidated sandy substrates in the lowlands (Figure 7). To characterize the acid-base status of forest soils, the base saturation can be classified in ecologically relevant groups of vertical gradients [59,68]. Here, the statistically defined six types by Hartmann and Wilpert [59] were used as a classification scheme.

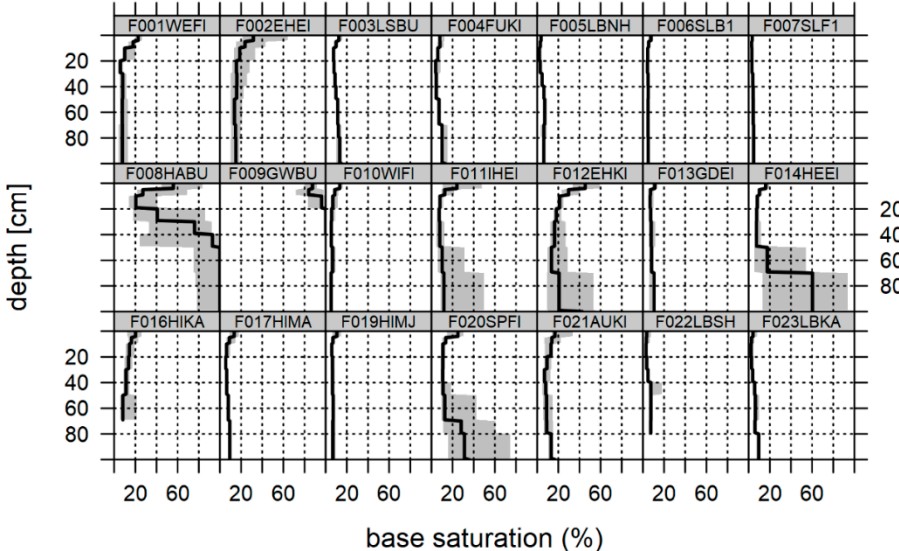

**Figure 6.** Depth profiles of pH(CaCl$_2$) in the mineral soil for the most recent soil inventory (cf. Figure 3) at the permanent soil monitoring plots in Lower Saxony. The solid line describes the median of six composite samples taken at 24 locations and the grey areas the error range (5th and 95th percentile).

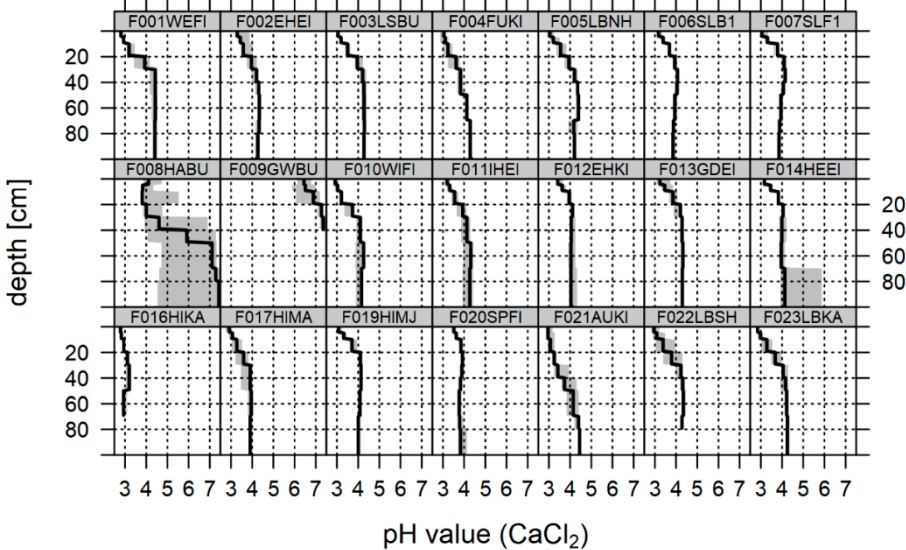

**Figure 7.** Depth profiles of base saturation in the mineral soil for the most recent soil inventory (cf. Figure 3) at the permanent study plots in Lower Saxony. The solid line describes the median of six composite samples taken at 24 locations and the grey areas the error range (5th and 95th percentile).

The groups 1, 2, and 3 are defined as soils with 100% base saturation in the (near) subsoil and very high (group 1—F009GWBU), high (group 2—F008HABU) or low (group 3) base saturation values in the topsoil. These sites are mainly found on limestone and some unconsolidated carbonate sediments. Group 4 is distinctly acidified in the main rooting zone with a high base saturation in the subsoil (F012EHKI, F014HEEI, and F020SPFI). All other plots can be assigned to group 5. The soils from this group are deeply acidified with very low base saturation values over the entire soil profile. Group 6 shows an increase of base saturation in the topsoil as a result of liming. The elevated base saturation in the upper 10 cm of the plot F012EHKI is typical for group 6. However, this plot was assigned to group 4 because of a high base saturation in the subsoil. Plots F011IHEI, F014HEEI, and F021AUKI were limed in the 1980s. However, the effects of liming were no longer evident

at the time of the last soil inventory. Accordingly, these limed plots were assigned to group 5, but not included in the comparison of forest types.

### 3.3. Statistical Detection of "Atypical" Monitoring Sites (Main Cluster Groups)

According to the principal component analysis of the soil chemical variables (pH, base saturation, and CEC) at the most recent soil inventory, three clusters of the permanent soil monitoring plots can be identified. Two clusters are represented by only one plot each, namely F008HABU and F009GWBU, respectively, whereas the third cluster comprise all other plots (Figure 8). According to this result, F008HABU and F009GWBU were excluded from the following analysis of the changes in soil pH and base saturation.

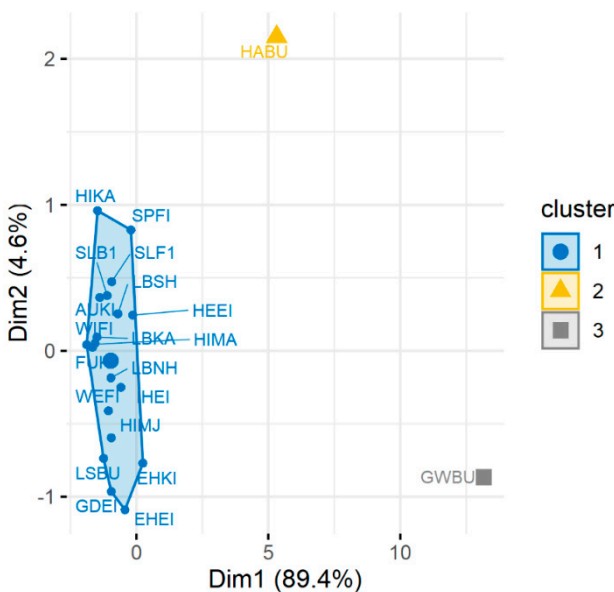

**Figure 8.** First two components of a principal component analysis of soil chemical variables at the most recent (Figure 3) soil inventory and designation of permanent soil monitoring plots to clusters 1 to 3. Clusters 2 and 3 consist of only one plot each. Abbreviations show the last four letters of the plot code (see Table 1).

### 3.4. Changes of the Acid-Base Status and Indications of Recovery

If all plots are evaluated, a significant decrease in base saturation up to about 1990 for all evaluation depths considered is visible (Figure 9). Neither significant increases nor decreases can be detected after 1990. The plots F021AUKI, F014HEEI, F011IHEI, and F012EHKI were limed between 1985 and 1992. At these plots, a significant increase in base saturation for all depth intervals occurred after liming (Figure 9; 'limed plots'). The unlimed plots showed no tendency toward recovery. Only if the deeper soil layers are included (0–100), a slight, but mostly insignificant increase of base saturation can be recognized. To delineate site and tree species effects, only the plots of base saturation group 5 were considered in the following evaluation (Figure 9; 'group 5'; 'group 5 coniferous'; 'group 5 deciduous'). All plots included in the tree species comparison have similar soil acid-base status on average. The depth-weighted average base saturations for the depth range 0–100 cm in the subgroups 'group 5', 'group 5 coniferous', and 'group 5 deciduous' are 7%, 7%, and 8%, respectively.

For the same groups, the depth-weighted average $pH(H_2O)$ values are 4.4, 4.4, and 4.5, respectively, and the $pH(CaCl_2)$ values are 4.0, 4.0, and 4.0, respectively. The effective cation exchange capacities are 402 $kmol_c\ ha^{-1}$, 422 $kmol_c\ ha^{-1}$, and 371 $kmol_c\ ha^{-1}$, respectively.

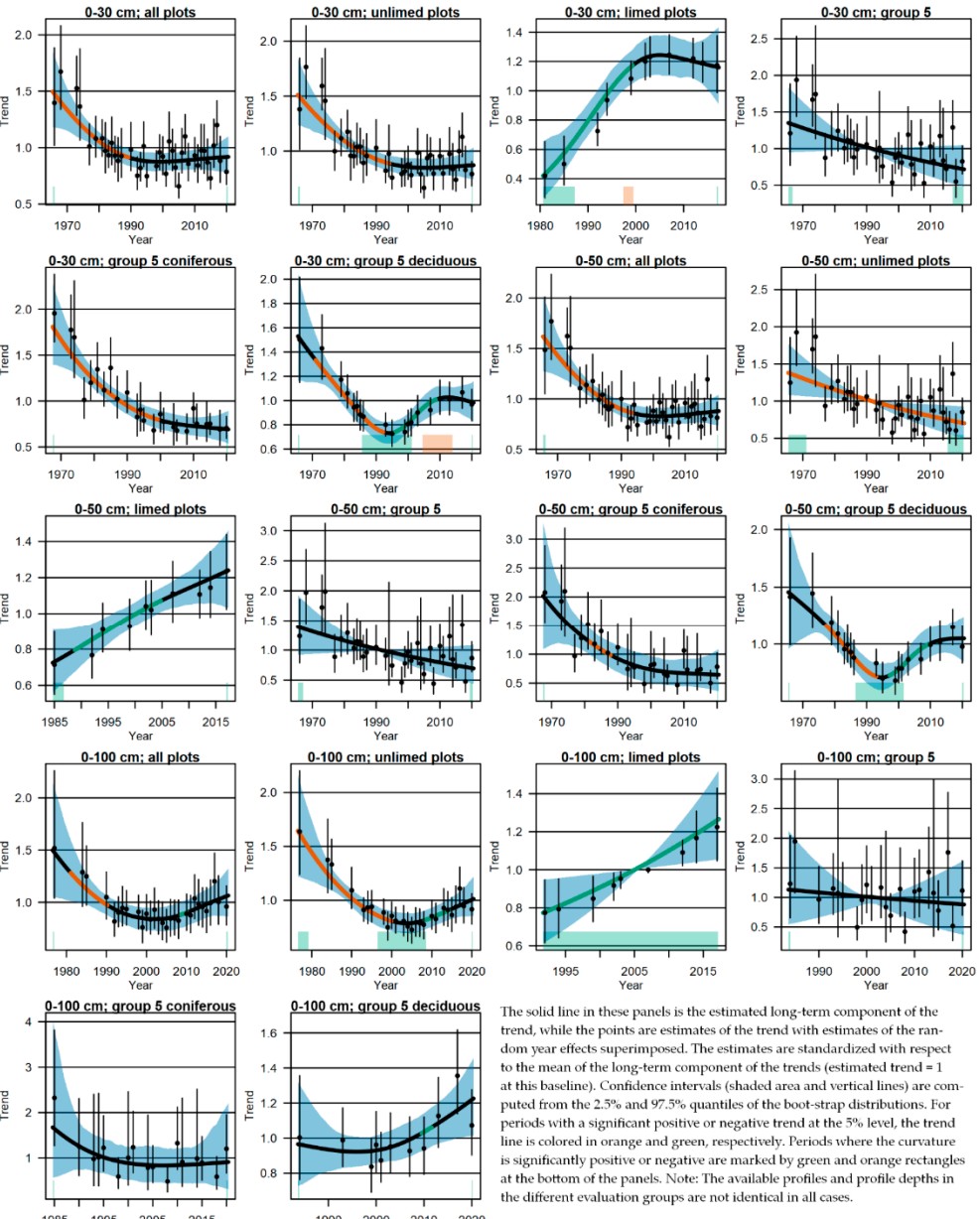

**Figure 9.** Estimated trends for the base saturation in mineral soil for different aggregation depths (0–30 cm, 0–50 cm, and 0–100 cm).

The solid line in these panels is the estimated long-term component of the trend, while the points are estimates of the trend with estimates of the random year effects superimposed. The estimates are standardized with respect to the mean of the long-term component of the trends (estimated trend = 1 at this baseline). Confidence intervals (shaded area and vertical lines) are computed from the 2.5% and 97.5% quantiles of the boot-strap distributions. For periods with a significant positive or negative trend at the 5% level, the trend line is colored in orange and green, respectively. Periods where the curvature is significantly positive or negative are marked by green and orange rectangles at the bottom of the panels. Note: The available profiles and profile depths in the different evaluation groups are not identical in all cases.

The difference in the dynamics between coniferous and deciduous forest sites is evident. At the deciduous forest sites, a period from 1990 to 2000 shows a significant positive curvature for the depth ranges 0–30 cm and 0–50 cm (Figure 9). A reversal of the decrease after 1990, with a significant increase of base saturation since 2010, is only visible for these two depth ranges. The lack of significance in the 0–100 cm depth range might be owing to the lower number of soil samples in this depth range. In contrast, a decrease of base saturation at coniferous plots continued until about 2000 in all depth ranges. In the subsequent period, the base saturation remained at a low level (<10%; Figure 7).

Observed pH(H$_2$O) is available from the early 1990s. As with base saturation, there has been a significant increase in pH(H$_2$O) at the plots with liming measures between 1985 and 1992 for all three depth ranges (Figure 10). There are indications of recovery at the unlimed plots between 2000 and 2010 (consistently significant for 0–50 cm and 0–100 cm). After this period, there appears to be a stabilization or even a slight decrease of the pH values.

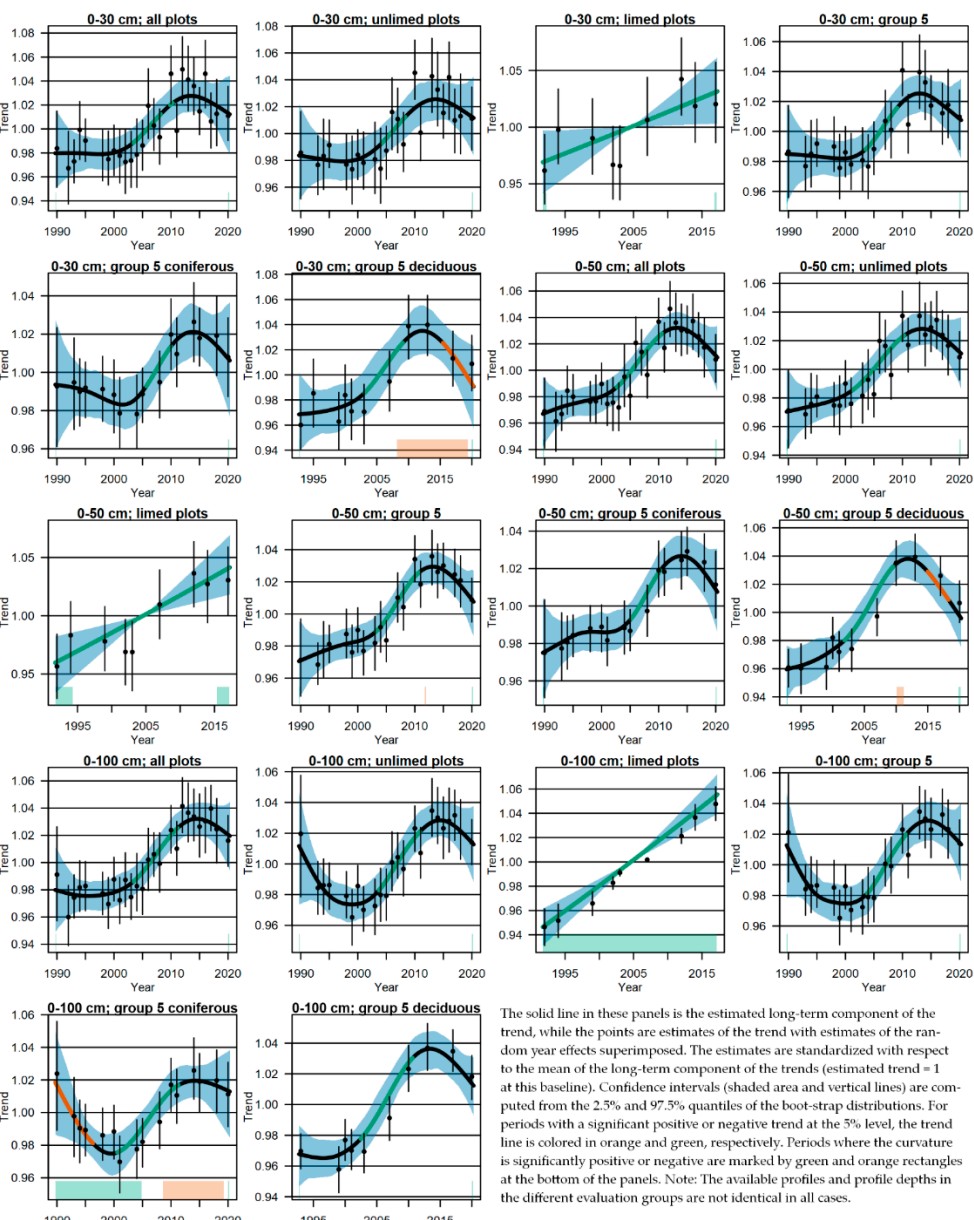

**Figure 10.** Estimated trends for pH(H₂O) values in mineral soil for different aggregation depths (0–30 cm, 0–50 cm, and 0–100 cm).

As compared to the deciduous forest plots, the recovery of pH(H₂O) was delayed at the coniferous plots, while the magnitude of the recovery is considerably more pronounced at the deciduous forests. This is most evident for the 0–50 cm depth interval. A decrease after 2010 should be interpreted with caution, as there are only inventories from two plots.

In contrast, the temporal patterns of pH(CaCl₂) are much more difficult to describe (Figure 11). The effects of liming do not appear to be as pronounced as for base saturation and pH(H₂O), especially in the top 30 cm. However, for the other two depth ranges (0–50 cm and 0–100 cm), there is a significant increase in pH(CaCl₂) after liming in the years 1985 and 1992. The temporal dynamics of pH(CaCl₂) show a very striking sinusoidal structure for the groups 'all plots' and 'unlimed plots'. These results show that interpretations are difficult when very different groups (soil types, parent material, and forest types) are analyzed together. When deciduous and coniferous plots from the deeply acidified group 5 are evaluated separately, this structure is no longer apparent. For pH(CaCl₂), there is no significant increase in the topsoil (depth ranges 0–30 cm and 0–50 cm), neither

at the deciduous nor coniferous plots. If the deeper soil horizons are included, there is a significant increase in the pH(CaCl₂) values, especially in the period from 2000 to 2010 for all aggregation levels of the subgroups ('all plots', 'unlimed plot', 'group 5', 'group 5 coniferous', and 'group 5 deciduous'). This is particularly interesting as sulfate deposition declined steeply until the year 2000 (Figure 2), but at a much slower rate afterward.

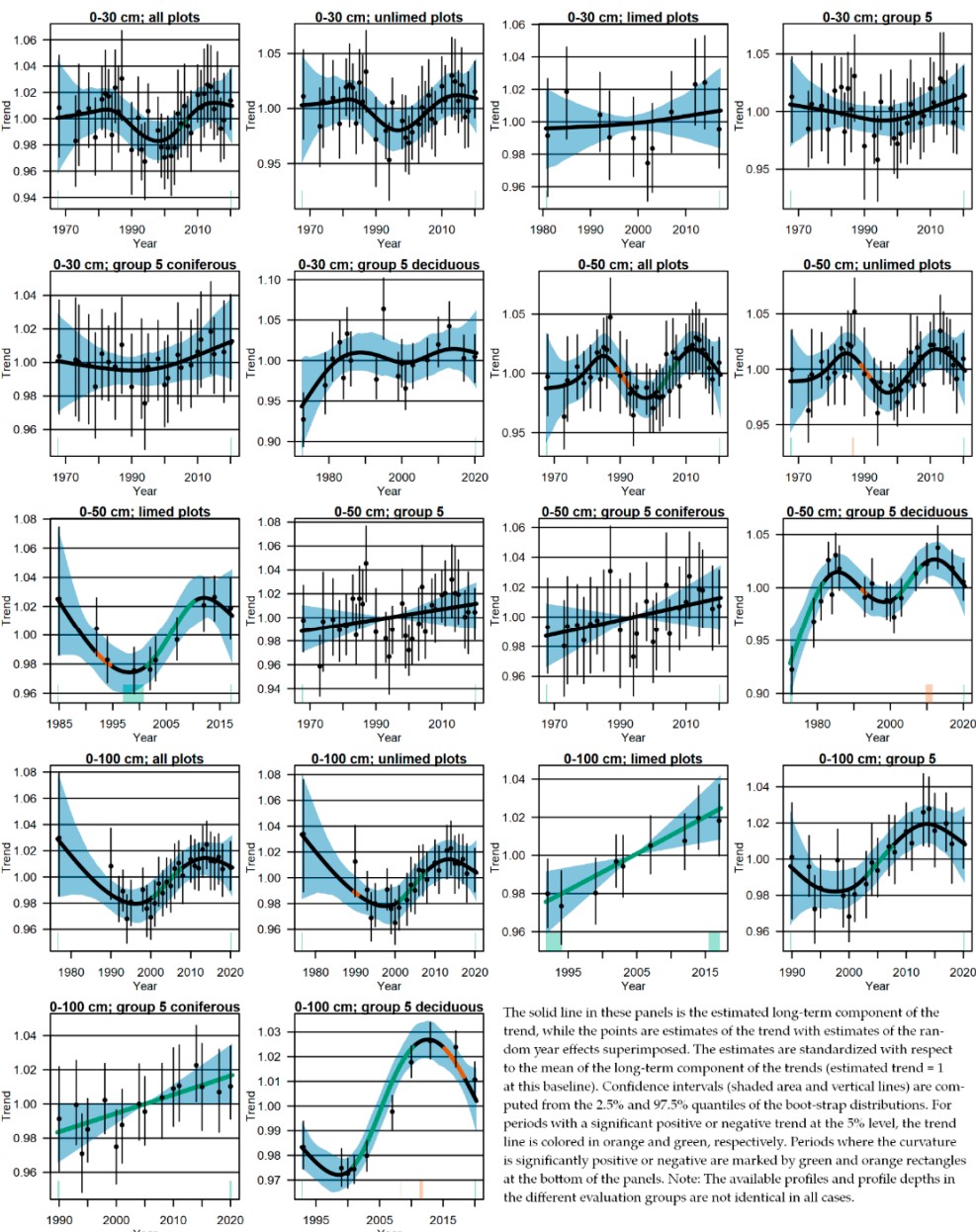

**Figure 11.** Estimated trends for pH(CaCl₂) values in mineral soil for different aggregation depths (0–30 cm, 0–50 cm, and 0–100 cm).

## 4. Discussion

### 4.1. Site-Specific Sulfur Deposition Time-Series

The approach for quantifying total sulfur deposition for the years 2000 to 2015 with an emissions-based method is described in detail in Schaap et al. [49]. In short, three major calculation steps are conducted in this model: (1) the chemical transport model LOTOS-EUROS [69,70] is used to calculate dry deposition as a product of modeled ambient air concentration fields of S species and modeled deposition velocities. (2) In the next step,

modeled rainwater concentrations from the LOTOS-EUROS model are used in combination with about 100 monitoring stations of precipitation chemistry in Germany. These data serve to adjust the modeled rainwater concentration distribution from the LOTOS-EUROS model using residual kriging. The generated concentration field is multiplied with high-resolution precipitation data (1 × 1 km). (3) Occult deposition is calculated from fog water concentrations. In such a model chain, there are numerous sources of uncertainties [9,71,72]. An example of the uncertainties caused by the spatial resolution of 1 × 1 km are the two sites at the Solling area (F006SLB1 and F007SLF1). While the deposition in the beech stand is well described by the modeling, considerable deviations from the observed deposition are shown for the neighboring spruce stand.

A further uncertainty arises from the reconstruction of the deposition with uniform deposition scaling functions for all sites. Despite these two main uncertainties, with the exception of site F007SLF1, there is a good agreement with observed deposition fluxes, both in terms of absolute magnitude and temporal dynamics (Figure 2). The highest S deposition rates were observed and estimated for coniferous forests (Figures 2 and 5).

The estimated S deposition for the soil inventory sites peaked in the early 1980s with maximum deposition rates of over 80 kg S ha$^{-1}$ a$^{-1}$. This is comparable with other European studies [73].

### 4.2. Soil Chemical Status, Sampling Design and Statistical Approach

A high proportion of deeply acidified soils in our study documents the strong acidification of forest soils in Lower Saxony. Very low pH values indicate a historically high acid load and the exhausted buffer capacities of the mineral soil. The second National Forest Soil Inventory (NFSI II) in Lower Saxony revealed a base saturation in the main rooting zone (10–30 cm) of below 20% at 65% of the forest area [74]. A base saturation of at least 15–20% is regarded as a critical limit for vital growth and the sufficient regeneration ability of most tree species [56,75]. If base saturation falls below this critical limit, a significant increase of toxic Al$^{3+}$ ions and an increasing mobilization of heavy metals into the soil solution can be suspected. While "natural" soil acidification from carbonic or organic acids is usually limited to the topsoil, most plots in this study experienced deep acidification, resulting from the transport of deposition-borne mobile anions such as sulfate and nitrate into deeper soil layers.

The detection of changes in the acid-base status of forest soils using repeated soil inventories necessitates a high accuracy of the inventories and sufficient changes of soil chemical variables between the inventories. Therefore, a high reproducibility of the sampling design and the comparability of the methods used for the repeated inventories is required [35,36,76]. Our data are arranged in a matrix of observations, in which some are missing because of the experimental design (Figure 3). Braun et al. [77] used linear mixed-effect models (LMM) with the plot as a random effect and the year as a fixed effect to analyze time trends in the soil solution. The inclusion of the plot as a random effect is necessary to account for the "pseudo-replicated" data structure (correlation among measurements on the same plot). Major challenges arise because the soil's chemical changes are often nonlinear and non-synchronous in the different soil horizons [31,36]. Additionally, soil changes are usually very slow and may be superimposed on the short-term and seasonal fluctuations [36]. Especially in forest soils, large spatial heterogeneities can contribute to a reduced detectability of soil changes [36,78]. In particular, the spatial variability of exchangeable base cations contents, which are needed for the calculation of base saturation may be very high [36,79], especially if the different depth intervals are considered [80].

The error ranges of the GAMMs reveal that the plot-scale variability is sufficiently captured given the sampling design (Figures 6 and 7). Since 1992, all study plots were sampled with an equal number of sampling points. For a few plots, the error range (in deeper soil layers) suggests that a higher number of replicates might be advisable, or even necessary. Furthermore, the ability to detect soil changes increases with an increased number of plots and sampling dates at the different plots. In our study, many plots have a

very high number of sampling dates over a period of 30 to 50 years (the median for all sites are four inventories; min = 2 and max = 12). An increasing number of sampling dates for a site can provide a more accurate assessment of soil changes over time. Mobley et al. [36] concluded that the sampling scheme must incorporate more sampling dates to better capture nonlinear structures in the dynamics of soil variables. We assume that in our study this advantage outweighs the disadvantages of a slightly different sampling design (lower number of repetitions) before 1992 (Figure 3). Soil inventories at the study plots prior to 1992 mainly document an advancing acidification process under high loads of acidic deposition. The course of a potential recovery from acidification because of a reduction of atmospheric acidic deposition during the period 1992–2020 is consistently substantiated through soil inventories with the same number of replicates in approximately 10-year intervals. During the whole study period, sampling procedures and analytical methods are checked by rigorous QA/QC routines, including the analyses of several hundred samples with new and old methods, in case a new analytical method was introduced, to secure the conformity of the methods [81].

The analysis of response variables with non-Gaussian distributions, the "pseudo replicated" structure of the inventories at specific sites, the consideration of non-linear change processes, the sampling heterogeneity over space and time, and the non-equidistance in the timing of the soil inventories, has been addressed in this study using GAMMs [43–45]. Knape [45] showed that the inclusion of temporal random effects in the estimation of smooth trends makes it possible to separate long-term changes from short-term fluctuations. In our study, we assume that short-term fluctuations are mainly caused by uncertainties in the observation of a site-specific representative mean of the different acidification indicators. The GAMM framework published by Knape [45] was developed for modeling trends in the count data of populations. Changing the response distribution to Gamma, the underlying pitfalls in modeling count data and long-term soil chemical variables are very similar. Soil data, as well as count data, could have a high variation in sampling effort and detectability. Therefore, a very flexible but robust statistical approach is needed. Such an approach is also a "classic GAMM". The main advance of the package 'poptrend' is the visual interpretability of the long-term recovery trend. A distinction can be made between periods with a significant increase, a significant decrease, and/or a stagnation of acidification indicators. A direct implementation of sulfur deposition in the modeling approach is a much greater challenge because the highest sulfur output occurs many years later than the occurrence of the highest sulfur deposition loads [21].

### 4.3. Change in Soil Chemical Variables and Indications of Recovery

After cluster analysis, two of our sampling sites were excluded from the study, as the soil chemical status was very different from the other sites and even from each other. This was also done against the following background: Kirk et al. [82] stated that in soils in the carbonate or silicate buffer range the pH values do not necessarily increase after a decline in deposition acidity [4]. A strong increase in pH values is more likely in the aluminum buffer range, where the dissolution of Al silicates, the destruction of clay minerals, and the protolysis of Al hydroxides are the primary soil chemical processes. Accordingly, Kirk et al. [82] found the largest increase in soils with low initial $pH(H_2O)$. In addition, other studies found a recovery in the most acidic forest soils ($pH(CaCl_2) \leq 4.0$ or BS $\leq 20\%$) [31,83].

Between 1970 and 1990, the $pH(CaCal_2)$ values and base saturation decreased in the soils of the studied permanent soil-monitoring plots. Hazlett et al. [20] summarized in their introduction results from numerous soil sampling studies across Europe and eastern North America. These resampling studies confirmed a decrease in the soil pH and base saturation in many forest soils for this period. Since the early 1980s, there has been a drastic reduction of the deposited sulfur and acidity in Europe and North America (cf. Figures 3 and 5). The comparison of 21 plots in Lower Saxony in this study showed a recovery from acidification indicated by an increase of soil pH and base saturation using repetitions of forest soil inventories. Particularly, the soil data from the Solling plots

(F006SLB1; F007SLF1) belong to the longest time series of repeated soil inventories in forest ecosystems worldwide. The observed and estimated sulfur deposition for the study plots shows a very different absolute reduction and cumulative load of sulfur deposition in the past (Figures 3 and 5). This is of great importance because some studies indicate that the intensity of the decrease in sulfur deposition could be linked directly to the degree of recovery [20].

Considering the group 'all plots', the base saturation for all aggregation depths (0–30 cm, 0–50 cm, and 0–100 cm) predominantly show a very slight increase, but the increase is not significant (Figure 9). Despite the considerable reductions in sulfur deposition in Lower Saxony (Figure 3), the possible recovery from soil acidification appears to be very slow. Meesenburg et al. [14] found for German NFSI plots that unlimed acid-sensitive sites experienced an ongoing acidification of the subsoil with corresponding losses of base cations. At liming trials in southwestern Germany, changes in untreated (unlimed) plots were only marginal and the soils remained highly acidic [84]. Major causes for a delayed recovery of forest soils from acidification are the still-substantial deposition of nitrogen species [1,16] with subsequent generation of acidity through uptake and nitrification processes, as well as the remobilization of previously retained sulfur over a longer period [29]. After a period of N accumulation with increasing nitrogen stocks, forest ecosystems may become N saturated, resulting in an increasing risk of leaching nitrate and base cations into surface waters [2]. In addition, changes in sulfur deposition were identified as key drivers in carbon stabilization and nitrogen leaching risks [3,85]. These processes, although seemingly linked to recovery, result in a significantly delayed recovery or even the further acidification of soils. Another reason for a very weak recovery is the simultaneous decrease in base cation deposition [86,87]. In our study, a trend reversal (insignificant) or stabilization at a low level can be observed at the depth intervals 0–30 cm and 0–100 cm. However, at the depth interval 0–50 cm an ongoing acidification seems to be evident. In the German NFSI, those sites, which were limed and are stocked with deciduous tree species, showed indications of recovery [14]. If the limed plots are examined individually in our study, there is a significant increase in base saturation after liming. It is well-known from numerous studies [14,84,88–90] that limed forest sites experience a significant increase in base saturation and pH values (especially in the topsoil). For example, Guckland et al. [88] found an 11% increase of base saturation up to a 40 cm depth. The German NFSI also found an increase of base saturation in the upper 30 cm of the limed plots [14].

All plots with vertical base saturation gradient type 5 were analyzed separately to reveal the potential effects of the tree functional groups (deciduous versus coniferous stands). While a recovery cannot be detected for the plots with conifers, the deciduous plots show a significant increase in base saturation after about 1995. The tree functional group-specific differences diminish with increasing soil depth. Figure 5 shows that the sulfur deposition load in the past was lower for deciduous than for coniferous forests. Accordingly, coniferous forest plots have a higher potential for the remobilization of temporarily stored sulfur. However, it should be noted that some of the coniferous permanent monitoring plots are located in highly polluted regions (Section 4.1). Therefore, the described tree functional group effect may to a certain extent be confounded by a plot-specific deposition load. Nevertheless, for the plots F006SLB1 and F007SLF1, it was demonstrated that the tree functional group effect is indeed decisive for the deposition input, retention of elements, and output fluxes [21]. However, it should be noted that this effect could not be solely because of lower deposition at deciduous stands compared to conifers. There are also many other factors and processes in the nutrient cycling and the related organic carbon dynamics of forest ecosystems that might explain differences in acidification dynamics between the forest function groups (conifers and deciduous) [22,91]. For example, litterfall, fine root turnover and decomposition, nitrogen retention, and depth and distribution of the rooting system may possibly differ significantly between the groups. Deep-rooting deciduous tree species [92] can enhance the zone, where mineral weathering contributes to the cycling of

base cations, and a higher base cation content of deciduous litter [93,94] may also have an effect on the replenishment of the cation exchange sites with base cations.

A consistent significant increase of $pH(CaCl_2)$ can only be observed in acidification depth gradient group 5 (Figure 11). Cools and De Vos [31] found that the $pH(CaCl_2)$ significantly increased at plots with very acidic forest soils ($pH(CaCl_2)$). For forest soils with $pH(CaCl_2)$ above 4.0, they found a further decrease. It should be noted here that there are very few plots with pH values above 4.0 in our study. When analyzing the different dynamics of soil recovery of deciduous and coniferous forests, the mean $pH(CaCl_2)$ values of the selected (see Figure 4 and Section 3.2) deciduous and coniferous plots are both 4.0.

The $pH(H_2O)$ shows a general increase after the drastic reduction of sulfur inputs since the early 1980s. However, the recovery takes place with a strong delay. In our study, the increase of $pH(H_2O)$ clearly appears later than the decrease in sulfur deposition. These results confirm other studies, according to which a significant delay can be assumed [20,22,82]. In our study, this time lag appears to be longer in the coniferous forests than in the deciduous ones. Watmough et al. [95] point out that the release of formerly stored sulfur delays the recovery from acidification. Other studies also show that a considerable proportion of formerly deposited sulfur is temporally stored in organic sulfur pools [21,96,97].

A consistent increase in soil pH was also found in Austria [98]. In Germany, the comparison between the first and second NFSI resulted in a significant increase of $pH(H_2O)$ in all depth intervals of the mineral soil [14]. Other studies from long-term soil monitoring programs in Europe, the United States, and Canada have shown that soil pH increases because of decreasing sulfur deposition [20,82]. In contrast, Berger et al. [29] found at the 'Vienna Wood' forest in the soil areas between trees and in the deeper soil horizons show no indications of recovery from acidification. Only within the stemflow soil area could a significant increase in $pH(H_2O)$ be found. They conclude that the recovery in the sampled soils in 1984 and again in 2012 at 97 beech stands may be highly delayed, especially in the deeper soil horizons. For the period between 1994 and 2007, an ongoing soil acidification was found in some parts of the Hrubý Jeseník region, Czech Republic [30]. In other parts, a slight decline in acidity was noted.

Our study shows different trends for $pH(H_2O)$ and $pH(CaCl_2)$. Meesenburg et al. [14] attribute this pattern to a reduction of the ionic strength in the soil solution, in particular due to the decrease in sulfur concentrations. At the time of the first soil inventories, many of our sites still had pH values at which the sorption of sulfur is particularly high [99]. Depending on the soil type, sulfur fixation often has its maximum at a $pH(H_2O)$ of 4.0 and remains approximately constant as the pH continues to decrease. However, if a reduced atmospheric sulfur load leads to a slight increase in pH and at the same time to a reduction in the sulfur content in the soil solution, the adsorbed sulfur is dissolved again and further acidification is promoted [100]. Thus, the reduction of acid inputs over the last three decades has only led to a significant increase of $pH(H_2O)$ in the soil solution. For $pH(CaCl_2)$, this development has not yet been clearly established.

Increases of base saturation and of $pH(CaCl_2)$ are only apparent for deciduous forests with lower historical sulfur inputs. The recovery of forest soils from acidification requires compensation for the very slow natural soil acidification [101] and other acidifying processes (e.g., timber logging and the natural accumulation of biomass) through weathering and other deacidifying processes. Different soil types with varying soil textures can have very different chemical weathering rates. Accordingly, the intensity of the acidification and recovery processes could be very different. For this reason, we divided the plots in groups with different vertical base saturation gradients and excluded the plots F008HABU and F009GWBU from the analysis.

Under certain conditions (low weathering rate, high utilization intensities), recovery is not to be expected at all in the decades to come [29]. The potential for a resupply of exchangeable base cations through weathering remains an important question because the uncertainty of the silicate mineral weathering rates estimates are very high [102,103]. Accordingly, some studies (summarized in [104]) concluded that it is unlikely that weath-

ering rates can replenish base cations to the extent necessary to bring about recovery. Sverdrup et al. [105] postulated a delayed recovery with an increasing soil depth. In contrast, in our study, there is no dampening of the change signals by adding deeper soil layers. In some cases, the signal becomes even clearer. Cools and De Vos [31] explained this development as follows: "In the acidification process, there could be a significant delay from the topsoil, which is first affected by acid deposition, to the bottom of the soil profile. During acidification, hydrogen and $Al^+$ ions mobilized in the soil solution may exchange with the base cations on a cation exchangeable complex and delay the decrease in pH. During recovery, the reverse process could occur, and while the upper layers recover, simultaneously the bottom layers may still acidify."

The significance of recovery for some subgroups (e.g., deciduous trees) shows that soil resampling appears to be a valuable method to detect soil changes over varying time periods at sites with different forest types and acid deposition histories. However, the lag time between the decrease in sulfur deposition and the recovery of soil chemical indicators underlines the importance of continuing long-term studies.

## 5. Conclusions

We conclude that despite a reduction of sulfur deposition by about 90% in Lower Saxony, the recovery from soil acidification is slow. The most recent soil inventories show a trend reversal or a stabilization at a low level. This recovery of the soils apparently occurred faster at deciduous compared to coniferous plots. A possible explanation for this finding could be a larger amount of temporarily stored sulfur in the soil because of higher atmospheric input in the coniferous forests. While the acidification indicators are still at a critical level and recovery is very slow and delayed in the coniferous forest soils, the acceleration of the regeneration process through liming still seems to be necessary. Furthermore, high nitrogen deposition loads in the Lower Saxony forests still appear to continue, resulting in an increasing risk of base cation and nitrate leaching into surface waters. Therefore, continued monitoring of the acid-base status of forest soils at permanent soil-monitoring plots seems to be necessary to track further ecosystem responses to changing environmental conditions, such as deposition, climate change, and weathering.

**Author Contributions:** Conceptualization, B.A. and H.M.; methodology, B.A., H.F. and H.M.; formal analysis, B.A. and H.F.; investigation, H.F. and H.M.; statistical analysis, B.A.; writing—original draft preparation, B.A.; writing—review and editing, B.A., H.F. and H.M.; visualization, B.A.; supervision, H.M.; project administration, H.M.; funding acquisition, H.M. All authors have read and agreed to the published version of the manuscript.

**Funding:** The study has been funded since 1992 by the State of Lower Saxony through the Permanent Soil Monitoring Programme. Partial funding of data collection and evaluation was provided by the European Union under Council Regulation (EEC) 3528/86 on the Protection of Forests against Atmospheric Pollution, the Regulation (EC) 2152/2003 concerning monitoring of forests and environmental interactions in the community (Forest Focus), and by the project LIFE 07 ENV/D/000218 Further Development and Implementation of an EU-level Forest Monitoring System (FutMon).

**Institutional Review Board Statement:** Not applicable.

**Informed Consent Statement:** Not applicable.

**Data Availability Statement:** The datasets related to this article are available from the corresponding authors on reasonable request. The original datasets for regionalized deposition are available on request from UBA, Dessau, Germany.

**Acknowledgments:** We thank our colleagues from the Northwest German Forest Research Institute involved in operating the network of permanent soil-monitoring plots in Lower Saxony in the last decades. The constructive comments by the four anonymous reviewers, which helped to improve the manuscript significantly, are gratefully acknowledged.

**Conflicts of Interest:** The authors declare no conflict of interest.

## Appendix A

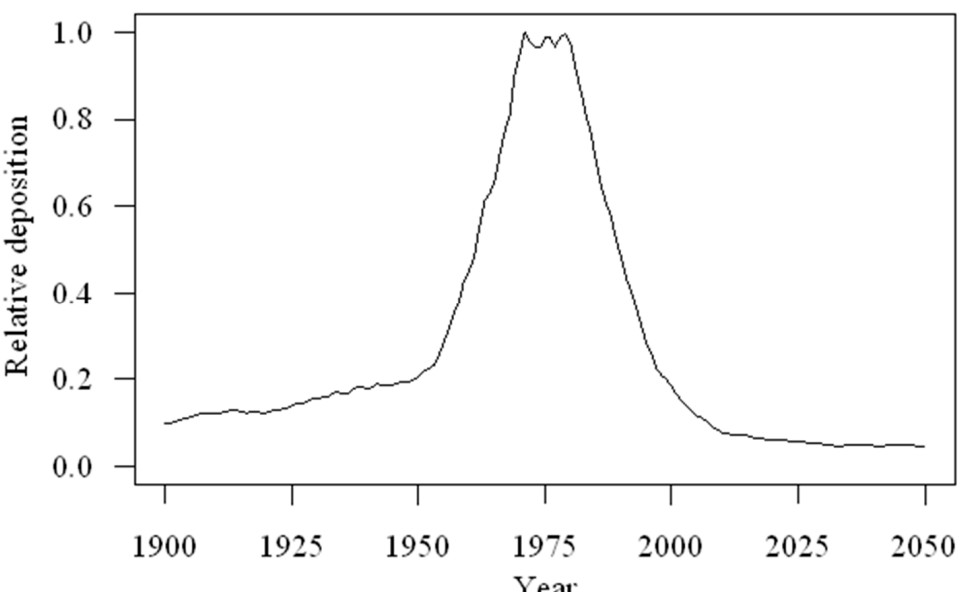

**Figure A1.** Standard curve for non-marine deposition in Lower Saxony, Germany. Adapted from Engardt et al. [9].

$$TD_{s(yr)} = \overline{TD}_{s(2000...2015)} \cdot SF_{(yr)} \cdot \overline{SF}_{(2000...2015)} \tag{A1}$$

where $TD_{s(yr)}$ is the annual sulfur deposition in the year $yr$ (kg ha$^{-1}$ yr$^{-1}$), $\overline{TD}_{s(2000\ldots 2015)}$ is the mean sulfur deposition for period from 2000 to 2015, $SF_{(yr)}$ is the annual specific scale factor in the year $yr$ (taken from Figure A1), and $\overline{SF}_{(2000\ldots 2015)}$ the mean scale factor for the period from 2000 to 2015.

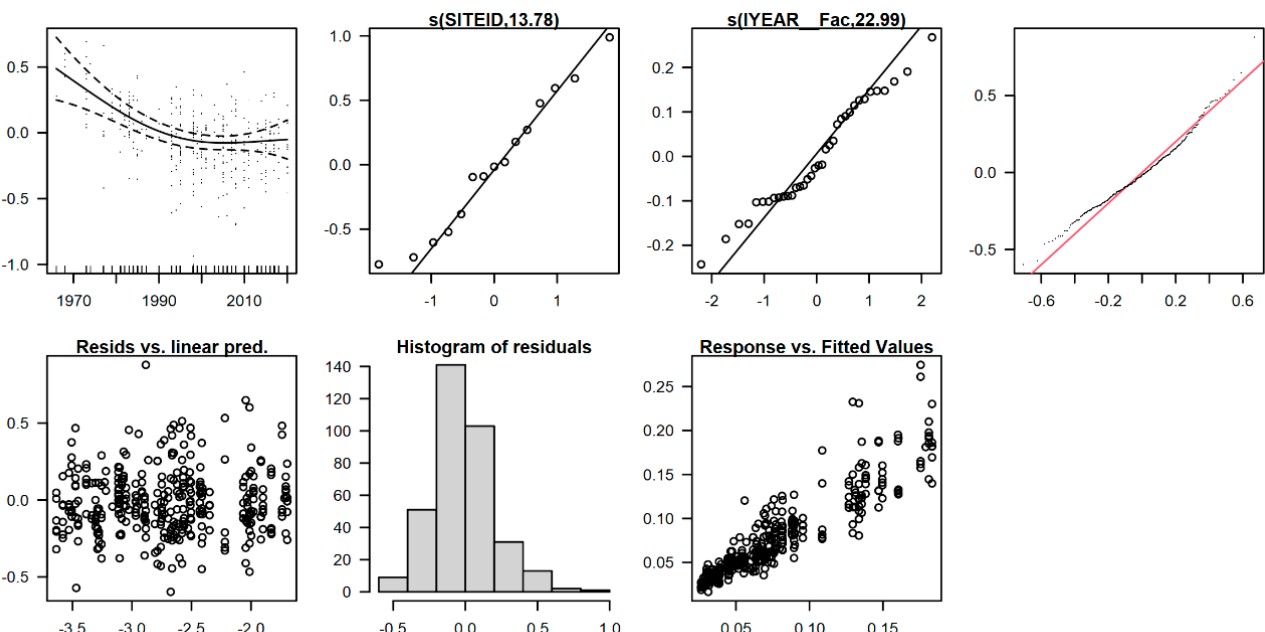

**Figure A2.** Example of graphical residual analysis (normality; homogeneity) of the selected generalized additive mixed model (GAMM) for the base saturation at unlimed plots in a 0–30 cm soil depth.

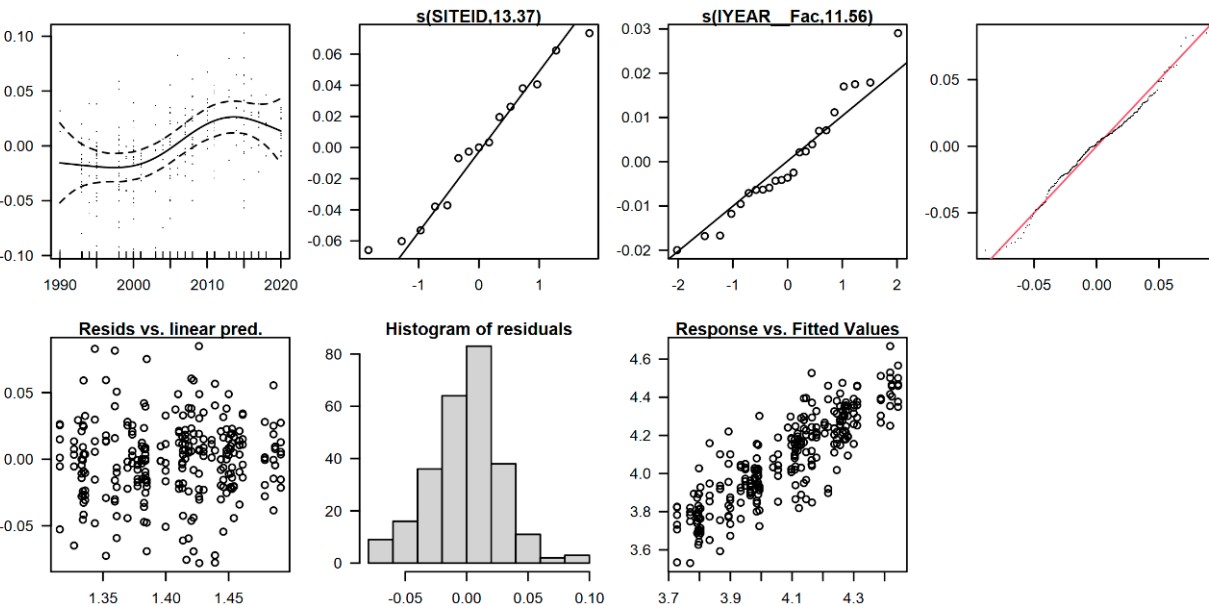

**Figure A3.** Example of graphical residual analysis (normality; homogeneity) of the selected generalized additive mixed model (GAMM) for pH(H$_2$O) values at unlimed plots in a 0–30 cm soil depth.

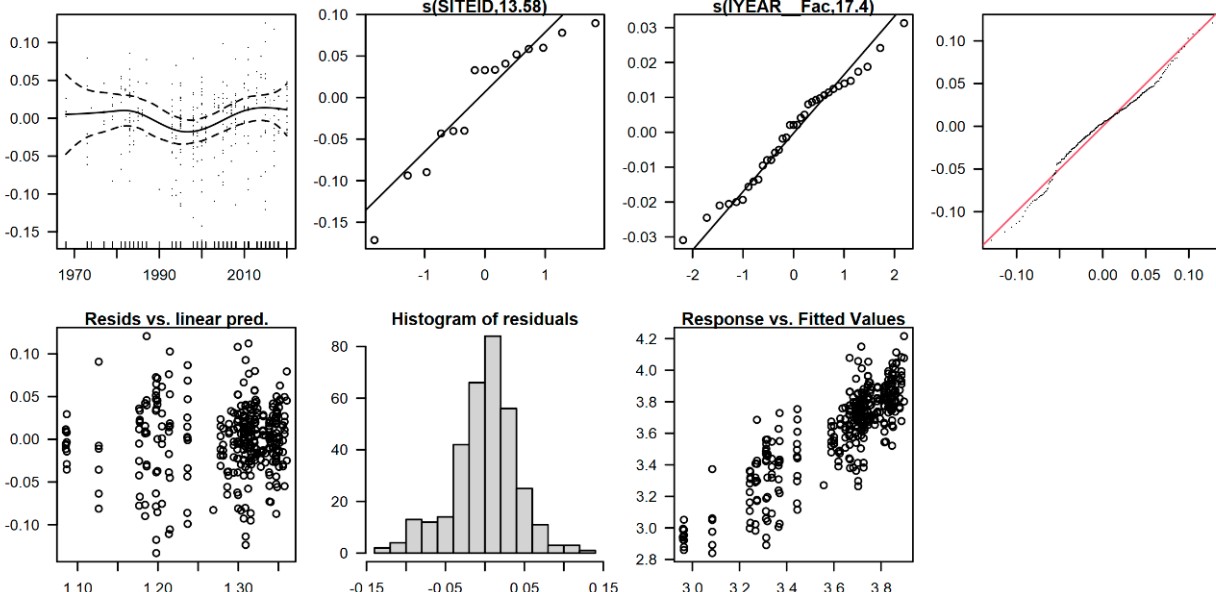

**Figure A4.** Example of graphical residual analysis (normality; homogeneity) of the selected generalized additive mixed model (GAMM) for pH(CaCl$_2$) values at unlimed plots in a 0–30 cm soil depth.

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
