# Peer review of "The Influence of Tree Species on the Recovery of Forest Soils from Acidification in Lower Saxony, Germany"

_soilsystems, doi:10.3390/soilsystems6020040_

Round 1

Reviewer 1 Report

This is a well written manuscript reporting an important long term study. A major contribution of this paper is the statistical method for handling data sets like this that seek to test trends despite problems of differences in sampling and spatial heterogeneity in soils.

The title and main conclusion emphasize the role of tree species in acidification and recovery processes. However, the rich selection of sites includes a wide variety of soil types, which are influenced by very different parent materials and soil forming processes. The roles of native soil processes in soil change could be better emphasized. Similarly, that the study plots include sites that have received liming mitigation is not emphasized. 

line 168: This should refer to figure 3?

line 180: Was H+ taken from the pH measurements? I don't believe it can be measured by ICP, although this is what the sentence says. And when it is determined from salt extraction, it is usually with a neutral salt such as KCl, not NH4Cl.

Figure 4: The label on the x axis of the right plot should read changes 1980 to 2020 (not the reverse).

Section 3.2 This paragraph is difficult to read and came as a surprise in the middle of the results. I suggest introducing the groups in the methods section and using a table to make it easier for the reader to compare between groups. 

Author Response

Response:

Dear reviewer, thank you for your time, suggestions and for the comments concerning our manuscript. Your comments are valuable for improving our research. We have made the following corrections which we hope will meet with approval.

We agree that studying sites with different soil types, parent materials and soil forming processes is a very problematic issue. To at least partially circumvent this problem, sites/plots were stratified into ecologically relevant groups of vertical gradients of base saturation before examining the role of tree species in acidification and recovery processes. Most of our study plots belong to the group of ‘deeply acidified soils’ (group 5 from Hartmann & van Wilpert 2016) with very low base saturation over the entire soil profile. We hypothesize that all of these plots are characterized by low weathering rates, a pedogenetic process that is critical for the long-term dynamics of acidification and recovery in unlimed forest soils. Accordingly, limed plots were excluded when examining the effects of forest type on recovery. The group of plots that were limed was additionally analyzed, because limed and non-limed plots are not comparable in their recovery dynamics.

To make all the listed points easier to understand for the reader, we have inserted Figure 4 (“Stratification of the inventory plots into functional groups for detailed statistical analysis”) in the manuscript and added the following sentence as an explanation to the Figure 4: “The plots were stratified into sub-groups to enable a more detailed analysis, as well as to consider the different soil chemical processes (such as chemical weathering rates) and other general conditions (such as liming), shown in Figure 4.” From this Figure, it can be seen that only the unlimed plots were used for the comparison between deciduous and coniferous forests. Additionally we separated the deeply acidified plots (with low weathering rates) from all other plots for the comparison of forest types.

We have also made the description of the results more detailed so that the reader has a better insight into what was compared with what and why. We have added a sentence to section 3.2 to clarify that limed sites are not included in the forest type-specific statistical analysis. The role of weathering as a native soil process is also now more clearly addressed in the discussion. We have included the following sentence: “Different soil types with varying soil textures can have very different chemical weathering rates. Accordingly, the intensity of acidification and recovery processes could be very different. For this reason, we divided the plots in groups with different vertical base saturation gradients and excluded the plots F008HABU and F009GWBU from the analysis.”

We mentioned in the discussion that soils with different pH values can react differently in the recovery process. To clarify for the reader that we only studied plots with comparable soil chemical condition to examine the effects of forest types, we have included the following sentence in the discussion: “When analyzing the different dynamics of soil recovery between deciduous and coniferous forests, the mean pH(CaCl2)-values from the selected (see Figure 4 and section 3.2) deciduous and coniferous plots are both 4.0.”

Some further recommendations from the reviewer

line 168: This should refer to figure 3?

Response: We have changed the reference to Figure 3.

line 180: Was H+ taken from the pH measurements? I don't believe it can be measured by ICP, although this is what the sentence says. And when it is determined by salt extraction, it is usually with a neutral salt such as KCl, not NH4Cl.

Response: Thank you! This was misleading. Exchangeable H+ concentrations were estimated from pH measurements prior and after percolation with NH4CL according to the given references. We changed the sentence to: “Exchangeable cations (Na+, K+, Mg2+,Ca2+, Al3+, Fe3+, Mn2+, H+) were determined after percolating sieved (< 2 mm) soil samples with NH4Cl and subsequent determination of cations in the percolate using ICP methods and pH measurements for H+ [53-55].

Figure 4: The label on the x axis of the right plot should read changes 1980 to 2020 (not the reverse).

Response: We agree and changed the label on the x axis of the right plot and inserted a new Figure 4. Note: recently now Figure 5

Section 3.2 This paragraph is difficult to read and came as a surprise in the middle of the results. I suggest introducing the groups in the methods section and using a table to make it easier for the reader to compare between groups. 

Response: We agree and introduce the groups and subgroups in the material and methods section. Instead of a table we inserted a Figure (Fig. 4) to provide a better overview of the groups and how they are formed.

Reviewer 2 Report

A brief summary

A review of the manuscript entitled: „The influence of tree species on the recovery of forest soils from acidification in Lower Saxony, Germany” is very interesting and promising. The results, conclusions and final discussion are sound. The manuscript is coherent and well written. It contributes to enhance the knowledge on this topic.

The data sets is good and nicely presented. Based on this general evaluation and the specific comments, reported below.  I recommend a minor revisions of the manuscript and re-write before it will be acceptable for publication. I have few specific comments, which might improve the manuscript.

Specific comment

Figure 7 – PCA analysis. Explain the abbreviations in the  PCA diagram. Probably are the abbreviations for plot code (see Table 1). Moreover, the division into three cluster is highly speculative. In my opinion PCA diagram showed one cluster and two single points =plots.

Line 360 – 367 this description belongs to methods.

Author Response

Response: Dear Reviewer, thank you for the positive comments which are very helpful for revising and improving our manuscript. We have made the following corrections which we hope will meet with approval.

Specific comment

Figure 7 – PCA analysis. Explain the abbreviations in the  PCA diagram. Probably are the abbreviations for plot code (see Table 1). Moreover, the division into three cluster is highly speculative. In my opinion PCA diagram showed one cluster and two single points =plots.

Response: Thanks, one explanation was missing: we added the following sentence in the caption of the Figure: "Abbreviations show the last four letters of the plot code (see Table 1)."
We agree that the term "cluster" might be a bit misleading at this point. You write: "In my opinion the PCA diagram showed one cluster and two single points = plots". However, this is exactly what we wanted to show with the graphical presentation of the PCA analysis. Based on this results, we excluded the two single plots from further analyses. In general, data clusters can be complex or simple, and a cluster is a subset of the entire data set, and each graph is closer to the center of the cluster than to the other cluster centers in the data set. Therefore, we believe that this is not a problem statistically. To clarify this, we have added the following sentence to the Figure caption: "Clusters 2 and 3 consist of only one plot each".

Line 360 – 367 this description belongs to methods.

Response: We agree and moved the part with the definition of the subgroups to the following subsection: 2.4 Data handling in the materials and methods section. We have included an additional Figure (Figure 4) to give the reader a better overview of the groups and subgroups. We also rewrote the beginning of subsection 3.4 (Line 360-367) a bit to put it back in context.

Reviewer 3 Report

Dear authors

The manuscript presents an interesting analysis of the dynamics of forest soils, with a multi-year approach helping to obtain a correct vision. The introduction is well structured and the discussion well developed, so in the methodology the discussion and justification part of the methodology used should be removed and simply explain the methodology applied in your work, moving the discussion part to the section where it should be included.  Thus, figure captions 8, 9, 10 have two font sizes

Also, in the methodology some scientific names are not in italics.

Best regards

Author Response

Response

Dear reviewer,

thank you for your time to read our manuscript and your positive feedback. We agree with your suggestion to move part of the statistical discussion to the discussion section. More specifically, we deleted line 245-251 from the Material and Methods section and incorporated some parts of those sections in the discussion, see section: 4.2 Soil chemical status, sampling design and statistical approach.

The two font sizes in the figure caption 8,9,10 were deliberately chosen by us, because the space for the extensive explanations was too small. To get around this, we completely restructured Figures 8-10 so that the explanation could be integrated directly into the figure. This makes Figures easier to read and the Figure captions consist of only one font size. Thank you for the suggestion.

Additionally, we have italicized all species names throughout the manuscript.

Reviewer 4 Report

The manuscript is generally well written. It contains clearly presented soil monitoring results that may be useful.

There are several points that make it difficult to fully understand the research. line 111: Why were these 21 plots used for the study? Were they selected or all available plots? 

line 118: Why is the reference period 1981-2010?

lines 119-125: Some of the references are too old. 

All Latin names of the species should be italicized.

The description and interpretation of the results could be more detailed. However, the discussion of the results is extensive. Both the Discussion and the Introduction include many relevant references.

The authors wrote the correct conclusions and indicated the need to continue the research. 

Author Response

Response:

Dear reviewer, thank you for your suggestions and comments concerning our manuscript. We have made the following corrections which we hope will meet with approval.

line 111: Why were these 21 plots used for the study? Were they selected or all available plots? 

Response: We used all forested plots with terrestrial soils from the permanent soil monitoring network in Lower Saxony for analysis. Grassland or cropland plots from the network were not considered in this study. To make this clear, we have re-written the sentence at line 111 and inserted an additional Figure 4 in the manuscript.

line 118: Why is the reference period 1981-2010?

Response: Until the end of 2020, the most current and widely used standard reference period was the 30-year period 1981-2010. We decided to calculate our climate data for this reference period, in order to standardize and harmonize our work with other studies.

lines 119-125: Some of the references are too old. 

Response: That was misleadingly done by us. Unfortunately, we have also used square brackets for the number of tree species in our study, as is the case with the references in the manuscript. To avoid such confusion, we have added an "n =" in each bracket in this section.

All Latin names of the species should be italicized.

Response: we have italicized all species names.

The description and interpretation of the results could be more detailed. However, the discussion of the results is extensive. Both the Discussion and the Introduction include many relevant references.

Response: We agree that the paper falls a bit short in describing the results. Accordingly, we have gone into much more detail in describing the results, particularly in Section 3.4.

The authors wrote the correct conclusions and indicated the need to continue the research. 

Response: Thank you!

Round 2

Reviewer 3 Report

Dear authors,

Thank you very much for your response to every required issue.

Best regards

Author Response

Thank you for your work!